# The key role of Au-substrate interactions in catalytic gold subnanoclusters

Jesús Cordón[1], Gonzalo Jiménez-Osés[1], José M. López-de-Luzuriaga [1] & Miguel Monge[1]

The development of gold catalysis has allowed significant levels of activity and complexity in organic synthesis. Recently, the use of very active small gold subnanoclusters (Au$_n$, $n < 10$) has been reported. The stabilization of such nanocatalysts to prevent self-aggregation represents a true challenge that has been partially remediated, for instance, by their immobilization in polymer matrices. Here, we describe the transient stabilization of very small gold subnanoclusters (Au$_n$, $n < 5$) by alkyl chains or aromatic groups appended to the reactive $\pi$ bond of simple alkynes. The superior performance toward Brønsted acid-free hydration of medium to long aliphatic alkynes (1-hexyne and 1-docecyne) and benzylacetylene with respect to phenylacetylene is demonstrated experimentally and investigated computationally. A cooperative network of dispersive Au···C–H and/or Au···$\pi$ interactions, supported by quantum mechanical calculations and time-resolved luminescence experiments, is proposed to be at the origin of this stabilization.

[1] Departamento de Química, Centro de Investigación en Síntesis Química, Universidad de La Rioja, Complejo Científico-Tecnológico, Madre de Dios 53, Logroño, La Rioja 26006, Spain. Correspondence and requests for materials should be addressed to G.J.-O. (email: gonzalo.jimenez@unirioja.es) or to J.M.L.-d.-L. (email: josemaria.lopez@unirioja.es)

The use of gold in catalysis has experienced a huge development in the last 25 years mainly due to its great catalytic performance under mild reaction conditions and its high selectivity in organic functional groups transformations[1–13]. Depending on the gold oxidation and aggregation states, different types of reactions have been developed. Thus, Au(0) in the form of small-size nanoparticles (Au NPs) are used in heterogeneous catalysis for the activation of C–X bonds (X=C, H, halogen, etc.)[10–13], while Au(I) and Au(III) species are widely employed as homogeneous catalysts for the activation of alkenes and alkynes[1–9].

Adding to the great number of ground-breaking applications described by the groups of Hashmi, Echavarren, Toste and others, Corma and co-workers have recently reported on a new type of an extremely efficient gold-catalyst consisting on ultra-small naked $Au_n$ clusters ($n = 3$–10 atoms), also called subnanometric gold clusters or gold subnanoclusters[14–17]. This type of small gold clusters can be prepared from either Au NPs or Au(I) or Au(III) species, which made the authors wonder about the identity of the real active catalysts in several of these homogeneous transformations. In one of these reports, the authors described the catalytic performance of solely $Au_3$–$Au_5$ subnanoclusters in the ester-assisted hydration of alkynes at very low catalyst loadings (parts per billion), or $Au_5$–$Au_9$ subnanoclusters in the bromination of arenes[14]. In the former case the catalytic activity strongly depends on the nature of the precatalyst, the formation and stabilization of the Au subnanoclusters being a critical and unresolved issue. Strikingly, the use of fatty additives bearing medium size alkyl chains led to a dramatic improvement of the catalytic properties of the gold subnanoclusters. This key finding, which was not discussed in the manuscript, suggests that the catalytic activity of $Au_n$ subnanoclusters might decrease due to competitive aggregation processes.

In view of these results, we anticipated that for this type of catalysis to be successful it is crucial that the reaction rate is higher than the self-aggregation of small $Au_n$ subnanoclusters. An adequate stabilization of the catalyst is necessary to maintain the smallest and most reactive $Au_n$ units separated; however, exceedingly strong interactions that would prevent catalyst turnover must be avoided. We envisioned dispersive, cooperative Au···$\pi$ and Au···H–C interactions between the substrate and the catalyst—such as those likely occurring in Corma's reaction with fatty esters—to be ideal for this purpose. The use of such bifunctional—reactive and protective—substrates would broaden the scope of this very promising technology, avoiding the use of surfactants, polymers or high-molecular weight solvents to stabilize the subnanoclusters.

Unlike in other transition metals[18], alkyl/aryl–gold interactions in Au(0) species are strikingly unexplored[19–21], particularly in catalysis. Most hydrogen bonds to gold atoms involve small, very polarized ligands[22], or biomolecules[23–29]. Dispersive Au···H–C interactions in $Au_n$ clusters are to date limited to computational predictions[30] lacking experimental validation. Also, although the existence of Au(I)/Au(III)···$\pi$ interactions is well documented[31, 32], aryl–gold interactions in $Au_n$ clusters are very scarce[33].

Here, we show through mechanistic, fluorescence, and computational studies that Au···H–C and Au···$\pi$ interactions between the catalyst and alkynes bearing flexible, medium-sized alkyl and aryl side chains preclude the aggregation and enhance the catalytic properties of small gold subnanoclusters.

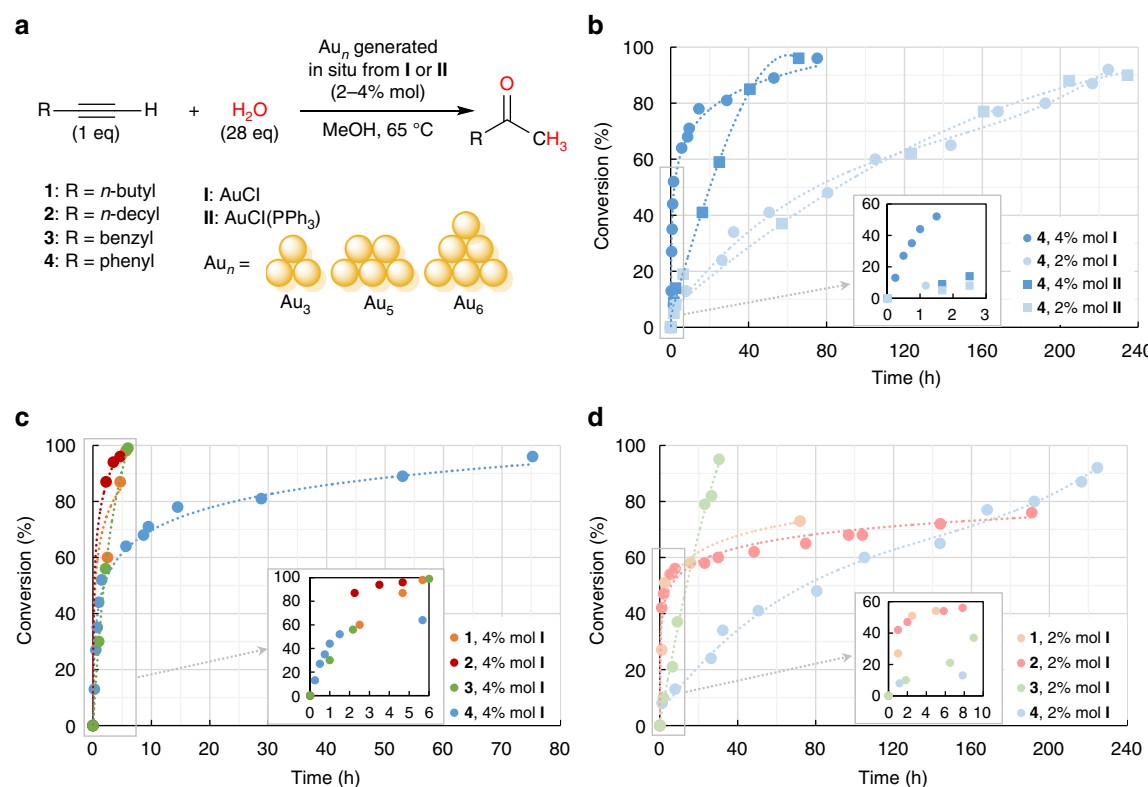

**Fig. 1** Hydration of alkynes catalyzed by Au(0) subnanoclusters. **a** Markovnikov hydration of alkynes in refluxing methanol catalyzed by Au(0) subnanoclusters of different sizes. **b** Kinetics of phenylacetylene (**4**) hydration catalyzed by Au(0) subnanoclusters generated from 4 and 2% mol of AuCl and AuCl(PPh)$_3$, showing that 4% mol AuCl is the most efficient precatalyst. **c**, **d** Kinetics of 1-hexyne (**1**) and 1-dodecyne (**2**) and benzylacetylene (**3**) hydration catalyzed by Au(0) subnanoclusters generated from 4 and 2% mol of AuCl, compared to those of phenylacetylene (**4**). A much higher reaction rate is observed for medium to long aliphatic and aromatic alkynes. The drastic loss of catalytic activity observed for aliphatic alkynes with 2% precatalyst after ca. ~3 h suggests decomposition of the Au(0) subnanoclusters

**Fig. 2** Scope of the Au$_n$-catalyzed alkyne hydration reaction. Alkenes bearing medium to long alkyl and aromatic side chains able to interact with gold subnanoclusters (**1**–**3**, **5**) react much faster than phenylacetylene derivatives (**4**) and (**6**). Shorter, more rigid side chains such as the tert-butyl group attached to (**7**) allow more discrete accelerations. The precatalyst loadings leading to highest conversion in each case are shown in parentheses

## Results

**Description**. We tested the possibility of enhancing catalysis through substrate-Au(0) dispersive interactions in the Brønsted acid-free Au$_n$-catalyzed hydration of 1-hexyne (**1**), 1-dodecyne (**2**) and benzylacetylene (**3**), using phenylacetylene (**4**) as a reference (Fig. 1a). Although phenylacetylene (**4**) contains an aryl substituent, it is directly attached to the coordinating linear alkyne and thus pointing outwards the subnanoclusters. Some flexibility in the alkyl/aryl side chain is envisaged to be necessary for an efficient recognition and stabilization of the gold subnanoclusters. Au$_n$ ($n = 3$–5) subnanoclusters were generated in situ from AuCl (**I**) or AuCl(PPh$_3$) (**II**) gold(I) precatalysts following previously reported procedures[14], and detected through matrix-assisted laser desorption/ionization-time-of-flight (MALDI-TOF) experiments. The generation of Au subnanoclusters from simple gold(I) precursors avoids the use of catalysts with more complex ligands, which results in an easier work-up protocol to carry out the catalytic process. Computational studies were performed to interpret the observed results.

**Kinetic assays**. In a first set of experiments the performance of Au$_n$ subnanoclusters toward the hydration of phenylacetylene (**4**) in methanol at 65 °C using **I** or **II** as precatalysts was studied (Fig. 1b and Supplementary Table 1). In all cases, formation of acetophenone was immediately observed under these relatively large precatalyst loading conditions, suggesting the very fast formation of catalytically competent Au$_n$ clusters; conversely, an induction period of several minutes had been previously observed for very low precatalyst concentration[14]. In order to check the possible formation of ketals prior to ketones through a hydro-alkoxylation reaction[34–36], we analyzed the composition of all reaction throughout time using gas chromatography-mass spectrometry (GC-MS). In previous reports, the formation of ketals was thoroughly studied both experimentally and theoretically. Under our conditions, only the hydration product was detected in all cases even at short reaction times. Both precatalysts provide high conversion levels at 4 mol% loading, albeit at modest reaction rates (>90% in ~70 h). However, it was apparent that 2 mol% of either **I** or **II** is insufficient to complete the hydration of phenylacetylene (**4**) at reasonable rates. The slower production of Au$_n$ clusters from AuCl(PPh$_3$) (**II**) with respect to the more efficient AuCl (**I**) was confirmed through $^{31}$P{$^1$H} NMR spectroscopy[14]. The slower production of subnanoclusters observed from the AuCl(PPh$_3$) precursor could be related to the slow formation of the real precatalyst AuCl through the equilibrium between AuCl(PPh$_3$) and AuCl and [Au(PPh$_3$)$_2$]Cl, as can be observed in the time-resolved $^{31}$P{$^1$H} NMR spectra (see

Supplementary Fig. 1). In agreement with our previous observations[37], the addition of HBF$_4$ (10% mol) led to the quantitative formation of acetophenone in 1 h (with **I**) or 7.8 h (with **II**). The formation of small size clusters (Au$_{1–5}$) was detected through MALDI-TOF mass spectrometry at the early stages of the reaction using 2% mol of **I** as the precatalyst (55 min), while larger clusters remained undetected. Larger nanoclusters up to Au$_{17}$ were observed at very long reaction times (262 h) (see Supplementary Figs. 3, 4). The influence of solvent in the catalytic hydration of phenylacetylene (**4**) was studied using 4% AuCl (**I**). Polar solvents such as tetrahydrofuran and acetonitrile led to poor conversions (4 and 14% in 53 and 48 h, respectively). In contrast, protic solvents such as methanol (96% in 75 h), ethanol (93% in 22 h), or 1-pentanol (91% in 53 h) provided almost quantitative conversions. Long-alkyl chain alcohols such as 1-decanol led to a decreased conversion (54% in 50 h), likely due to the poor solubility of water in this solvent and/or to the competition with the alkyl alkynes for the Au cluster, although the higher stabilization of the Au$_n$ species by the long-alkyl chain would have been expected.

The dramatic effect of the alkyne side chain on the reaction rate was demonstrated by using medium to long aliphatic (1-hexyne (**1**) and 1-dodecyne (**2**)) as well as aryl-substituted alkynes (benzylacetylene (**3**)) (Fig. 1c, d and Supplementary Table 2). To our delight, no induction period was observed and the reactions were completed in less than 6 h for all the substrates using 4 mol % AuCl (**I**) as a precatalyst. These results indicate a more than ten-fold acceleration with respect to phenylacetylene (**4**), and demonstrate the highly beneficial effects of appending a flexible alkyl/aryl side chain to the triple bond. Of note, at lower precatalyst loads (2% mol) the reactions with 1-hexyne (**1**) and 1-dodecyne (**2**) slow down dramatically after 3 h, and do not show the slow but steady progression observed for phenylacetylene (**4**) (Fig. 1c). These observations confirm that aliphatic alkynes are intrinsically poorly reactive, and that Au$_n$ subnanoclusters are deactivated after a certain period of time, likely through aggregation processes as detected by MALDI-TOF (see Supplementary Figs. 3, 4) and observed visually. A brownish solution typical of ultrasmall gold nanoparticles (<2 nm) together with a small amount of a purple solid are observed after 1 h of reaction with phenylacetylene (**4**). Long reaction times lead in all cases to the formation of a black precipitate corresponding to bulk gold.

In an attempt to rule out potential hydroalkoxylation reactions taking place at short reaction times, yielding transient ketals that would ultimately hydrolyzed to the observed ketones, long-chain alcohols such as 1-decanol in methanol were tested as nucleophiles in the presence of benzylacetylene and 4% mol AuCl as a catalyst. Under these conditions, no hydroxyalkylation

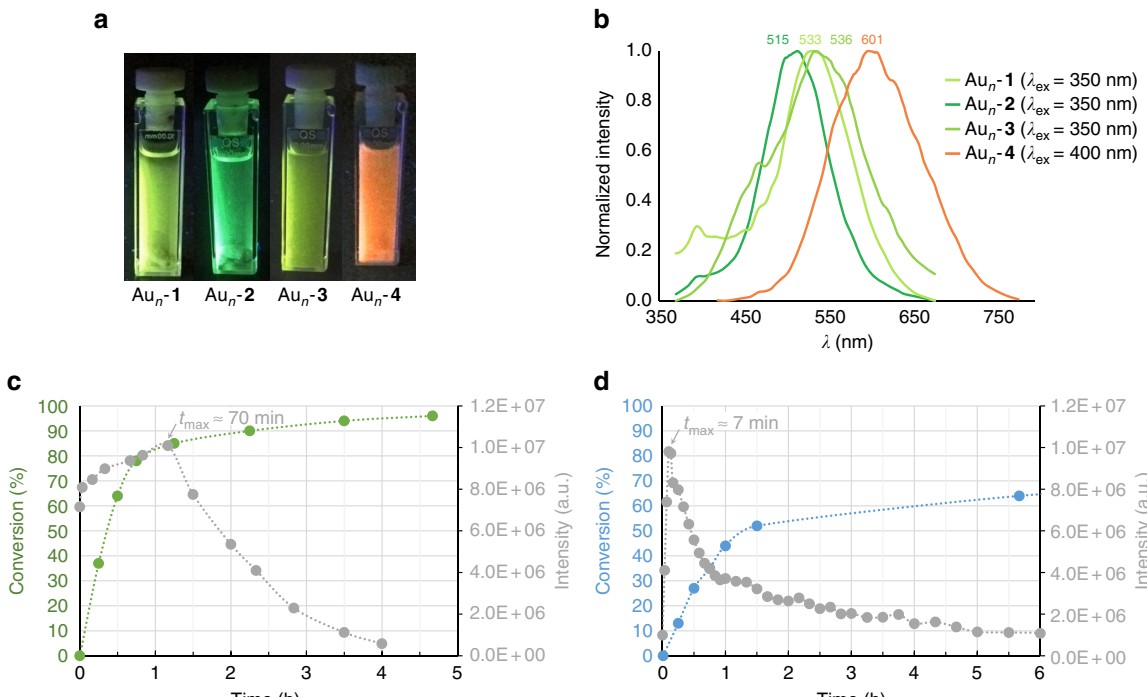

**Fig. 3** Emissive properties of Au_n-alkynes complexes. **a** Pictures of reaction mixtures prepared from alkynes (**1–4**) and AuCl (**I**) as a precatalyst, obtained under UV lamp irradiation (365 nm). The species formed with 1-hexyne (**1**), 1-dodecyne (**2**), and benzylacetylene (**3**) emit at higher energy (green-yellow) than phenylacetylene (**4**) (orange), indicating the formation of smaller size Au subnanoclusters with alkynes 1-3. **b** Emission spectra acquired for these same mixtures under irradiation at 350–400 nm. **c, d** Time-resolved emission intensity (in gray) vs. alkyne conversion in the catalytic hydration of 1-dodecyne (**2**, in green) and phenylacetylene (**4**, in blue). A longer high emission intensity (~70 min) was measured for (**2**) compared to (**4**) (~7 min). Conversion rapidly increases during the high-intensity emission intervals, and decreases when emission decays. All experiments were performed using 4% mol AuCl (**I**) as a precatalyst in methanol at 65 °C

products could be detected by GC-MS at any reaction time, and only trace amounts of the corresponding hydration product were observed due to the presence of traces of water in methanol. However, ketals were observed when 1,2-diols were used as nucleophiles. As expected, the reaction between benzylacetylene and ethylene glycol in the presence of 4% mol of AuCl led to the quantitative formation of the corresponding cyclic ketal in 5 h. The scope of the Au_n-catalyzed alkyne hydration[37, 38] was evaluated by testing substrates with different substitution patterns (Fig. 2). The observed results clearly demonstrated, as predicted, that alkynes bearing linear, flexible alkyl or aryl side chains able to interact with the gold subnanoclusters, such as 1-hexyne (**1**), 1-dodecyne (**2**), benzylacetylene (**3**) and 1-phenyl-4-pentyne (**5**) (>90% conv in 5–8 h), were more proficient than those lacking this ability such as phenylacetylene (**4**) and 1-ethynyl-4-pentylbenzene (**6**), both requiring over 50 h to achieve decent conversion levels. Branched, shorter alkynes such as 3,3-dimethyl-1-butyne (**7**) showed an intermediate reactivity (92% conv in 19 h), indicating a moderate capacity to stabilize small gold subnanoclusters.

Besides water and alcohols, other nucleophiles such as thiols and amines were tested. Addition of these nucleophiles to the tested alkynes was not observed in any case. Instead, peaks corresponding to Au(I) species were observed in the MALDI spectra of the reaction mixtures. With propanethiol, species of general formula $[Au_n(propanethiolate)_{n+1}]^-$ were formed due to the higher acidity of thiols compared to alcohols, which precluded the formation of subnanoclusters. Similarly, peaks corresponding to $[Au_n(1\text{-hexyne-H})_{n-1}]^+$ ($n = 6$, 7, 8, and 9) species were observed in the MALDI + spectrum when propylamine was used as a nucleophile; thus, under this conditions amines are able to

deprotonate alkynes, again precluding the formation of Au_n subnanoclusters (see Supplementary Fig. 2).

**Photoluminescence studies.** Since Au_n-alkyne species could not be isolated, we attempted to spectroscopically characterize the different stages of the reaction, taking advantage of the size-dependent luminescent properties of polymer or ligand-stabilized gold subnanoclusters[39]. Zheng et al.[40] described the emission of poly(amidoamine)-encapsulated gold subnanoclusters of different sizes: Au_5 (385 nm, UV), Au_8 (456 nm, blue), Au_13 (510 nm, green), Au_23 (751 nm, red), and Au_31 (879 nm, near IR). Jin et al.[41] synthesized thiolate protected Au_3 subnanoclusters showing an emission at 340 nm in the UV region. Following the same trend, Gonzalez et al.[42] described poly(N-vinylpyrrolidone)-stabilized Au_2 and Au_3 subnanoclusters emitting at 315 and 335 nm, respectively. Besides particle size, other factors potentially affecting the emission energies in these systems could be the ligand(s) and/or the solvent interacting with the subnanoclusters. Solvent effects, despite having an influence of the energy of the excited states, can be safely ruled out in our case because all the experiments are carried out in the same solvent; we assume that solvent effects are independent of the subnanocluster size. The influence of the alkyne ligands on the molecular orbitals involved in the electronic transition responsible for the observed luminescence was evaluated through density functional theory (DFT) calculations on model systems [Au_3(1-hexyne)] and [Au_6(1-hexyne)_2]. A very small contribution from the alkyne ligand compared to that from the Au_n subnanocluster was observed on the computed HOMO and LUMO orbitals (see Supplementary Fig. 5). Therefore, we can conclude that the coordinated alkynes

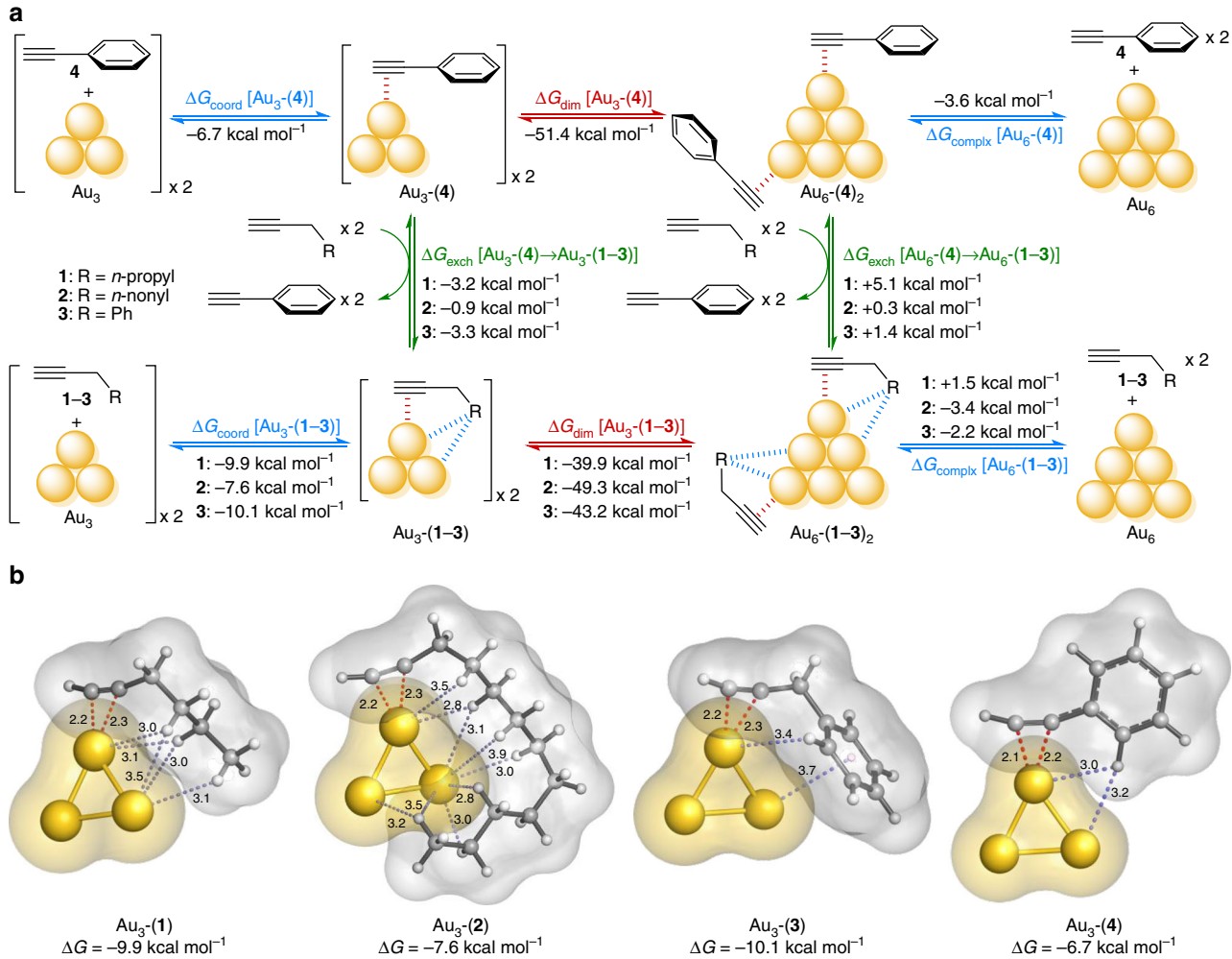

**Fig. 4** Quantum mechanical studies of alkyne coordination to gold subnanoclusters. **a** Calculated thermodynamic cycle for alkyne coordination and Au$_3$ dimerization. The different coordination abilities of phenylacetylene (**4**) and linear alkynes 1-hexyne (**1**), 1-docecyne (**2**) and benzylacetylene (**3**) to Au$_3$ and Au$_6$ subnanoclusters are estimated based on the calculated free energies for each equilibrium ($\Delta G_{coord}$, in blue). The preference of each nanocluster (Au$_3$ or Au$_6$) to bind the different alkynes is calculated through the corresponding exchange free energies ($\Delta G_{exch}$, in green). The ability of each alkyne to stabilize small (Au$_3$) or bigger (Au$_6$) subnanoclusters is derived from the calculated dimerization free energies ($\Delta G_{dim}$, in red). **b** Minimum energy structures and van der Waals surfaces calculated for Au$_3$-(**1-4**) complexes. Interatomic distances between the gold atoms and alkyne (in red) and side chain atoms (in gray) are shown as dashed lines and given in angstrom. Only Au···C–H and Au···$\pi$ contacts below 4 Å are shown. See Supplementary Fig. 9, Supplementary Table 3, and Supplementary Methods for further details

may have some, although not decisive, influence on the emission energy of the Au$_n$(alkyne)$_m$ species. In view of these results, it is expected that smaller Au$_n$ clusters show blue-shifted emissions.

The emission spectra for the Au$_n$-alkyne species with 1-hexyne (**1**), 1-dodecyne (**2**), benzylacetylene (**3**), and phenylacetylene (**4**) were registered shortly after mixing the reactants using 4 mol% of AuCl (**I**) as a precatalyst and at different reaction times (Fig. 3a, b). For compounds (**1-3**) the emission wavelengths of the corresponding Au$_n$-alkyne species are very similar with maxima between 515 and 536 nm. In clear contrast, the emissive behavior of the Au$_n$-**4** species is clearly red-shifted (maximum at 601 nm). This is the first evidence of alkynes bearing medium to long alkyl/ aryl side chains stabilizing smaller, more reactive gold sub-nanoclusters. Note that since in our case the gold nanoclusters are complexed to alkynes, the observed wavelengths are not directly comparable to the values described above, so no structural information about the size of the detected species can be inferred from this study.

The different extent of catalyst stabilization achieved by alkynes (**1-4**) was further evidenced by their time-resolved luminescence profiles acquired throughout the reaction, which reflect notable differences in the rates of generation and subsequent disappearance of the catalytically active species. Thus, emission for 1-dodecyne (**2**) steadily increases for ~70 min, nearly the time required for the reaction to complete (Fig. 3c). In contrast, emission for phenylacetylene (**4**) quickly decays after only ~7 min, when barely 10% of conversion has been achieved (Fig. 3d).

In fact, a remarkable correlation between time-resolved luminescence and rate of alkyne hydration was observed: the emission intensities—and thus the concentration of catalytically relevant Au$_n$-alkyne species—increase precisely while the reaction rates are high; after reaching its maximum, luminescence decays due to catalyst aggregation and consequently the reactions slow down dramatically.

In summary, time-resolved luminescence studies showed that alkynes bearing medium to long alkyl/aryl side chains not only stabilize smaller, more reactive Au$_n$-alkyne intermediates than phenylacetylene (**4**), but also extend their persistence in solution, allowing a much faster reaction completion.

**Quantum mechanical calculations**. The superior performance of medium to long aliphatic and aryl-substituted alkynes (**1–3**) vs. phenylacetylene (**4**) in the $Au_n$-catalyzed hydration reaction was studied computationally using DFT methods (see "Computational details" in the "Methods" section). First, the whole reaction pathway was calculated for phenylacetylene as a substrate and $Au_3$ or $Au_5$ subnanoclusters as catalysts, based on previously proposed mechanism using Au(I) and Au(III) species as catalysts[36, 37, 43, 44] (see Supplementary Figs. 6, 7).

Through this study it was found that unsolvated $Au_3$ is the most catalytically active species compared to methanol-coordinated $Au_3$ and unsolvated $Au_5$, leading to an energy span-energy difference between the turnover limiting intermediate and transition state of ca. 21 kcal mol$^{-1}$ consistent with the observed reaction times.

The whole free energy profile for the $Au_3$-catalyzed hydration of the experimentally most reactive alkyne, i.e., 1-hexyne (**1**) was subsequently calculated (Supplementary Fig. 8). While the enol-to-keto tautomerization (ts5) and protodeauration steps (ts7) have similar activation barriers than those calculated for phenylacetylene (**4**), the initial nucleophilic attack (ts3) to $Au_3$-(**1**) has a higher activation barrier than to $Au_3$-(**4**), due to the higher stability of the $Au_3$ complex with 1-hexyne (**1**) and the poorer electrophilic character of 1-hexyne (**1**) compared to phenylacetylene (**4**). These results strongly suggest that the formation and stability of the smallest Au subnanoclusters is indeed the catalytic bottleneck for this process, given that a kinetically less reactive alkyne such as 1-hexyne (**1**) is in fact the one that shows the highest hydration rates.

Once $Au_3$ was established as a convenient computational model for the type of highly reactive $Au_n$ subnanoclusters that are generated under the reaction conditions, we investigated the ability of different alkynes to coordinate to such subnanoparticles and preclude their aggregation. $Au_3$ and $Au_6$ subnanoclusters were chosen as convenient models to describe a prototypical aggregation process (i.e., dimerization). Figure 4a shows the calculated thermodynamic cycles accounting for these competing processes in solution. Coordination energies ($\Delta G_{coord}$) are used to quantify the relative strength of the complexes of $Au_3$ or $Au_6$ with 1-hexyne (**1**), 1-docecyne (**2**), benzylacetylene (**3**), or phenylacetylene (**4**) (denoted as $Au_{3/6}$-(**1–4**)).

Similar stability trends can be derived from isodesmic substitution reactions between these complexes, which are characterized by exchange free energies ($\Delta G_{exch}$). Finally, aggregation of the catalytically active $Au_3$ complexes ($Au_3$-(**1–4**)) to their less-reactive $Au_6$ counterparts ($Au_6$-(**1–4**)) is described by dimerization free energies ($\Delta G_{dim}$).

As depicted in Fig. 4a, complexation of 1-hexyne (**1**), 1-dodecyne (**2**), and benzylacetylene (**3**) to $Au_3$ ($\Delta G_{coord}[Au_3$-(**1–3**)] = −9.9, −7.6, and −10.1 kcal mol$^{-1}$, respectively) is favored with respect to phenylacetylene ($\Delta G_{coord}[Au_3$-(**4**)] = −6.7 kcal mol$^{-1}$), due to the presence of several Au···H–C and Au···$\pi$ interactions in the former (Fig. 4b). Note that $Au_3$-(**2**), despite achieving a larger number of Au···H–C interactions, shows a smaller interaction energy than $Au_3$-(**1**) and Au3-(**3**). This is due to the conformational penalty ($\Delta G_{folded-linear}$ = +2.9 kcal mol$^{-1}$) experienced by 1-dodecyne (**2**) upon folding into the conformation that is recognized by $Au_3$; such conformational penalty is negligible for less flexible 1-hexyne (**1**) ($\Delta G_{folded-linear}$ = +0.0 kcal mol$^{-1}$) (see Supplementary Table 4). Binding of all alkynes to $Au_6$ is disfavored with respect to $Au_3$, likely due to the loss of conformational entropy upon binding two alkyne molecules; this is particularly relevant for 1-hexyne (**1**) for which coordination becomes endergonic ($\Delta G_{coord}[Au_6$-(**1**)] = +1.5 kcal mol$^{-1}$). The same trend is reflected by the calculated isodesmic alkyne exchange reactions: $Au_3$ clusters prefer to bind 1-hexyne (**1**)

rather than phenylacetylene (**4**) ($\Delta G_{exch}[Au_3$-(**4**)→$Au_3$-(**1**)] = −3.2 kcal mol$^{-1}$), while more rigid phenylacetylene is a much better ligand for $Au_6$ ($\Delta G_{exch}[Au_6$-(**4**)→$Au_6$-(**1**)] = +5.1 kcal mol$^{-1}$). These results demonstrate the stronger preference for alkyl alkynes to stay bound to small, catalytically active subnanoclusters rather than to less active bigger particles.

Although aggregation was calculated to be a very thermodynamically favored process, as revealed by the large stability of $Au_6$-(alkyne)$_2$ species compared to their $Au_3$-alkyne counterparts, we found that 1-hexyne (**1**) largely decreases dimerization free energies by more than 11 kcal mol$^{-1}$ with respect to phenylacetylene (**4**) ($\Delta G_{dim}[Au_3$-(**1**)] = −39.9 kcal mol$^{-1}$ vs. $\Delta G_{dim}[Au_3$-(**4**)] = −51.4 kcal mol$^{-1}$). The same trend was calculated for 1-dodecyne (**2**) ($\Delta G_{dim}[Au_3$-(**2**)] = −49.3 kcal mol$^{-1}$) and benzylacetylene (**3**) ($\Delta G_{dim}[Au_3$-(**3**)] = −43.2 kcal mol$^{-1}$). Following the Bell–Evans–Polanyi principle for such exergonic reactions, alkynes (**1–3**) not only disfavor but very likely also slow down the aggregation of $Au_3$ to bigger particles, as observed experimentally through time-resolved luminescence (Fig. 3b, c).

The role of solvent on the relative stability of $Au_3$-alkyne complexes was examined (Supplementary Fig. 9). Irrespective of the nature (methanol or water) or number of solvent molecules coordinated to $Au_3$, the general trend calculated for the unsolvated $Au_3$ clusters is maintained: complexes with phenylacetylene (**1**), which lacks a flexible side chain able to interact with the metal cluster, are always less stable than those with 1-hexyne (**1**), 1-dodecyne (**2**), and benzylacetylene (**3**).

Finally, the superior stability of the $Au_3$-(**1**) complexes toward aggregation is reflected by the fact that we could not find any transition state or intermediate for the linearization of the $Au_3$ cluster starting from this adduct, which could be envisioned as a prerequisite for the formation of larger clusters such as $Au_6$. Any attempt to linearize $Au_3$ in $Au_3$-(**1**) quickly rearranged back to the triangular ground state, reflecting the protecting properties of alkynes bearing medium to long alkyl/aryl side chains. However, such $Au_3$ linearization transition state and intermediates could be indeed found for $Au_3$-(**4**) (Supplementary Table 4), suggesting its higher propensity to achieve higher nuclearity clusters.

## Discussion

Our joint experimental and computational study has established the proof-of-concept that medium-sized alkyl and aryl side chains appended to reactive $\pi$ bonds, selectively and transiently stabilize the smallest, most reactive Au subnanoclusters through well-balanced Au···H–C and Au···$\pi$ attractions between the catalyst and the substrate. This approach provides an effective way to outcompete catalyst aggregation while retaining its high reactivity toward the desired reaction, eliminating the need of exogenous matrices and nanoparticle stabilizers. Transferability of this approach to other hydroaddition reactions and different metal subnanocatalysts will be examined in the near future.

## Methods

**General procedures**. Compound AuCl(PPh$_3$) was synthesized according to published procedures[45]. AuCl was acquired from Aldrich. MALDI-TOF spectra of subnanoclusters were recorded in a Microflex MALDI-TOF Bruker spectrometer. $^1$H and $^{31}$P{$^1$H} NMR spectra were recorded on a Bruker AVANCE 400 instrument in deuterated methanol. Chemical shifts are quoted relative (external) to H$_3$PO$_4$ for $^{31}$P. The quantitative monitoring of reaction was performed by gas chromatography using a Hewlett-Packard G1800B GCD system, equipped with a Teknokroma TRB-1 cross-linked dimethylpolysiloxane column (30 m × 0.25 mm × 0.25 μm) and MS detector (electron impact with single quadrupole filter). A split injection system with a split ratio of 50:1 was used with helium as carrier gas at head pressure of 16 psi. Temperature programming was 80 °C (2 min), 20 °C/min, 240 °C (10 min). The inlet temperature was 225 °C and the detector temperature was 250 °C. Conversion of the starting material and product yield were measured by integrating the chromatographic peaks of phenylacetylene (retention time 2.51 min) and acetophenone (retention time 4.17 min). No internal or external standard

was used since both compounds showed a similar response factor ($K_{acetophenone}/K_{phenylacetylene} = 1.02$). Emission spectra were recorded on a Jobin-Yvon Horiba Fluorolog 3-22 Tau-3 spectrofluorimeter. Measurements at 65 °C were done using a Julabo F25-MV circulating bath accessory connected to the cuvette sample holder.

**Typical catalytic experiment**. The corresponding gold pre-catalyst (2–4 mol%) was placed in a round-bottomed flask. Methanol (5 mL), water (500 µL), and the corresponding alkyne (1 mmol) were sequentially added, and the mixture was magnetically stirred at 65 °C (methanol reflux) for the time considered in each case. Then, to analyze the type of subnanoclusters present and the yield of the catalysis, a sample was taken at each time for MALDI-TOF and GC-MS analysis, respectively. The MALDI-TOF samples were prepared by adding 1 µL of the reaction mixture on a spot of a MSP ground steel BC sample holder (Bruker). The samples were air dried before measurement.

**Computational details**. All geometry optimizations were carried out using the M06-2X hybrid functional[46]. In all calculations, the heteroatoms were treated by SDD pseudopotentials[47], including only the valence electrons for each atom. For these atoms, double-zeta basis sets were used, augmented with d-type polarization functions[48]. For H atoms, a double-zeta basis set was used, together with a p-type polarization function[49]. The 19-valence electron SDD pseudopotential[50] was employed for Au atoms, together with two f-type polarization functions[51]. A benchmark study of different DFT functionals was carried out prior to the use of M06-2X for the whole computational analysis. B3LYP[52–54], Grimme's B3LYP-D3[55], Head-Gordon's ωB97xD[56], and Truhlar's M06[46] and M06-2X[46] functionals were tested in the full optimization of the wrapped conformation of 1-hexyne-Au$_3$ (see Supplementary Table 3 and Fig. 10). Briefly, with B3LYP we could not locate local minima showing the alkyl side chain wrapping around the Au$_3$ cluster. In contrast, Grimme's B3LYP-D3, Head-Gordon's ωB97xD, and Truhlar's M06 and M06-2X did allow full optimization of such folded structures, which resulted to be more stable than the extended ones. We selected M06-2X for the whole study because it has proved to be a robust and cost-efficient method for organic/organometallic chemistry, especially for addition reactions to multiple bonds[57]. Full geometry optimizations and transition structure (TS) searches were carried out with the Gaussian 09 package (Frisch, M. J. et al., Gaussian, Inc., 2009).[58] The possibility of different conformations was taken into account for all structures. Frequency analyses were carried out at the same level used in the geometry optimizations, and the nature of the stationary points was determined in each case according to the appropriate number of negative eigenvalues of the Hessian matrix. The quasi-harmonic approximation reported by Truhlar et al.[59] was used to replace the harmonic oscillator approximation for the calculation of the vibrational contribution to enthalpy and entropy. Scaled frequencies were not considered. Mass-weighted intrinsic reaction coordinate calculations were carried out by using the Gonzalez and Schlegel scheme[60, 61] in order to ensure that the TSs indeed connected the appropriate reactants and products. Bulk solvent effects were considered implicitly by performing single-point energy calculations on the gas-phase optimized geometries, through the IEFPCM polarizable continuum model[62] as implemented in Gaussian 09. Gibbs free energies ($\Delta G$) were used for the discussion on the relative stabilities of the considered structures. Au$_3$ and Au$_6$ subnanoclusters were modeled as a radical doublet and a closed-shell singlet, respectively. Diverse contributions (electrostatic, exchange, repulsion, polarization, and dispersion) to the interaction energies were calculated for selected model Au$_3$ adducts through the localized molecular orbital energy decomposition analysis[63] as implemented in Gamess[64]. This analysis revealed that while Au$_3$-acetylene, Au$_3$-benzene, and Au$_3$-water interactions are clearly dominated by electrostatic/polarization attractions, in the case of Au$_3$-butane dispersion is equally important to electrostatic/polarization (Supplementary Fig. 11). Cartesian coordinates, electronic energies, entropies, enthalpies, Gibbs free energies, and lowest frequencies of the different conformations of the low-energy conformations are available in Supplementary Table 4 and below.

**Data availability**. All data are available from the authors upon reasonable request.

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

## Acknowledgements

D.G.I. MINECO/FEDER (project numbers CTQ2016-75816-C2-2-P (AEI/FEDER, UE) to J.M.L.L. and CTQ2015-70524-R(AEI/FEDER, UE) and RYC-2013-14706 to G.J.-O.) is acknowledged for financial support. J.C. also acknowledges MEC for a FPU grant. We gratefully thank CESGA, UR (Beronia cluster), and BIFI (Memento cluster) for computer support.

## Author contributions

J.M.L.-d.-L., M.M. and G.J.-O. designed the study. J.C. performed all experiments and J.C. and M.M. performed the calculations. J.C., G.J.-O., J.M.L.-d.-L. and M.M. analyzed the data and wrote the manuscript.

## Additional information

**Competing interests:** The authors declare no competing financial interests.

