## [Peer Review File · Nature Communications]

Reviewer #1 (Remarks to the Author):

This is a well-balanced study combining experiments, spectroscopic studies and a detailed computational study on very small gold clusters.

These sub-nanoparticles are a hot topic in catalysis research and enable very high turnover numbers.

The principle of stabilization of the sub-nanoparticles by an interaction of the alkyl side chains of substrates by dispersive interactions is a highly important one.

the manuscript clearly should be published in Nature communications.

Before the publication the following points have to be addressed:

"Recently, the use of exceedingly active small gold subnanoclusters (Au_n , $n < 10$) has been reported by the group of Corma (Science 338, 1452–1455 (2012))."

-- extend this sentence to

Instead of reference (1)

a specific review article should be provided, I would strongly suggest

"while Au(I) and Au(III) species are widely employed as homogeneous catalysts for the activation of alkenes and alkynes.1–8"

-- extend this to

"while Au(I) and Au(III) species are widely employed as homogeneous catalysts for the activation of alkenes, allenes and alkynes.1–8"

and add the key citation for allenes: Angew. Chem. Int. Ed. Engl. 2000, 39, 2285-2288.

"Most hydrogen bonds to gold atoms involve small, very polarized ligands^{19,20}"

-- here add a citation of Ito's pioneering milestone paper after polarized ligands:

J. Am. Chem. Soc. 1986, 108, 6405-6406

"also called subnanometric gold clusters or gold subnanoclusters.13–16 "

-- here as a reference add the detailed discussion of this success of Corma:

Science 2012, 338, 1434-1434.

"The scope of the Au_n -catalysed alkyne hydration"

-- after the word "hydratation" add a references:

"For high turnover numbers in the related spiroketalization of alkynes, see: Angew. Chem. Int. Ed. 2013, 52, 7963–7966"

"based on our previously proposed mechanism using Au(III) species as catalysts³⁸ (see Supp. Inf. Figs 3 and 4)."

-- here add two more references on details of these activations of alkynes by gold(III):

Schwerdtfeger's study in Organometallics 2010, 29, 2206–2210 and Pernpointner's fundamental relativistic study in J. Chem. Theory Computation 2009, 5, 2717–2725.

After these additions, the manuscript is ready

Reviewer #2 (Remarks to the Author):

This manuscript reports the preferential formation, stabilization and, incidentally, catalysis of ultrasmall Au clusters (<5 atoms) in the presence of long-chain alkyl alkynes. This phenomenon explains the different catalytic activity found for AuCl and AuPPh₃Cl as precursors of Au clusters

during the hydration of different alkynes: while alkyl alkynes react well, phenylacetylene react sluggishly, a paradigmatic shift respect what should be expected (aromatic alkynes are more prone to nucleophilic addition) and also respect the well-known reactivity of phenylacetylene reported in the literature. Experimental and computational studies here support that the increasing number of Au—H-C dispersive interactions between the methylene groups of the alkyl chain of the alkyne and the ultrasmall Au clusters is the reason of their stabilization, and that when the cluster is >5 Au atoms, entropy takes over and more than one alkyne coordinates, thus Au—H-C interactions mainly disappear.

This is a foundational work that opens new ways of selectively when reacting alkynes. It seems plausible that the findings here shown are not only applicable to the hydration reaction but also to other hydroaddition reactions and other metal clusters, at least those of late heavy transition metals able to interact well with C-H bonds (Pd, Pt...). The manuscript is well-written and easy-to-follow, and the results seem easy to reproduce by other laboratories. Thus, I recommend publication after the following issues are addressed:

- Other nucleophiles rather than water: What about diols, amines and thiols? It would be nice to have some preliminary results, which would greatly improve the impact of the work.
- In alcohol media, it is probable that ketals are formed prior to the ketone (hydroalkoxylation reaction). Authors claim do not find ketals by GC-MS even at short reaction times, however, non-cyclic ketals will not be stable under the reaction conditions used. To detect if ketals are intermediates of the reaction, two possible methods can be employed: 1) Direct detection with a glycol as nucleophile, which forms the stable 5-member cyclic ketal; 2) As the alkyl chain increases, steric hindrance decreases the ketal formation rate, so the order of reactivity should be methanol>ethanol> isopentanol \approx pentanol>> decanol. This comparison must be done with the initial rate of formation of the ketone (the product is the same for all the alcohols) taken from kinetics; in the manuscript, authors refer to quantitative conversions in the alcohols but at final reaction time. A hydroalkoxylation reaction as first reaction step would explain the decreasing activity in 1-decanol.

Reviewer #3 (Remarks to the Author):

The paper by Cordon et al. addresses the role of dispersive interactions in stabilizing small gold species in solution and the catalytic activity of small gold clusters in hydration reactions. The reviewer's comments are listed below.

1. The comparison of the performance of the catalyst needs additional support.

A) The overall kinetics plotted in Figure 1b for I Au and II Au shows very similar time dependency for the 2% samples. In the case of the 4% samples there are fewer data points shown for the II Au sample - this sample reaches \sim 95% conversion faster than the 4% I Au sample. Could the authors comment in more details and update the plots with error bars, if possible?

B) In order to further fortify the authors' claim the 4% I Au catalyst being the most active one, it would be very helpful if the authors could also include kinetics data obtained with the 2% and 4% II Au catalyst in reaction with some of the compounds (1) to (4).

2. The description of the NMR results shown in Supplemental Figure 1 should be expanded:

A) There is an apparent discrepancy: The figure caption mentions monitoring $[\text{Au}(\text{PPh}_3)_2]\text{Cl}$ and AuCl , however the peaks in the plot are labeled as $[\text{Au}(\text{PPh}_3)_2]^+$ and $[\text{ClAu}(\text{PPh}_3)_2]$.

B) The authors should also include a discussion of the evolution of the features with time and how this supports their claim made in the main text (page 5) about different rates of cluster formation when using different precursors.

3. A faster formation of Au clusters after the addition of HBF_4 is claimed through change of the reaction rate. Could the authors include direct evidence or a literature reference on the accelerated cluster growth under these conditions?

4. MALDI-TOF is used to determine the size of the clusters (Supplemental Figure 2), which is a very elegant method to determine particle size.

A) In the caption of FigS2 Au₃ is described as the most abundant species at early reaction times, while the main text refers to the abundance of 1-5 atom clusters. For clarity, can the authors assign all peaks in the spectra?

B) For a better assessment of the size of clusters present at early and late times, could the authors expand both spectra for showing a range corresponding to cluster sizes ~1-20?

5. Page 5 of the main text refers to solvent effects to be listed in Supplementary Table 2. However, the table lists kinetics data for different substrates which are referred to on page 6 of the manuscript.

6. Photoluminescence is used to estimate cluster sizes, based on literature references.

A) This reviewer is not a specialist and would like to learn from the paper how to distinguish size effects from ligand/solvent effects. Could the authors discuss this point in the manuscript?

B) The time resolved emission spectra show a very nice correlation between catalytic conversion and abundance of the species emitting at the given wavelength. Did the authors also collect emission data at other wavelengths to monitor the presence of different size clusters?

Other comments:

- This reviewer also suggests adding more details on MALDI characterization in the Methods section, including sample handling.

- The font size is very small in some of the figures.

In summary, Cordon et al. amassed an impressive set of data. However, in the opinion of this reviewer, a more focused presentation would have been better suited for a short communication. This reviewer feels that this work is better positioned for a full paper in a more specialized journal.

Reviewer #4 (Remarks to the Author):

The work may be in principle of interest for the wide chemists community working in homogeneous and heterogeneous catalysis.

The authors claim that the transient stabilization of very small gold subnanoparticle (for example Au₃) can be achieved appending alkyl chains or aromatic group to the coordinating π bond of simple alkynes.

The authors tended to correlate the superior thermodynamic stability of small gold cluster with alkynes bearing alkyl chains or aromatic groups with experimental kinetic-profiles for alkynes hydration reaction.

This correlation is interesting but, in my view, authors do not appropriately discuss their results in the context of previous literature.

The catalyst stability is clearly a key point, however pre-equilibrium, nucleophilic addition, and protodeauration are as much as important and the final outcomes depend on each of these steps.

Several works in the literature have already pointed out most of these aspects just as in the case of hydroalkoxylation of alkynes mediated by gold.

The interpretative framework emerging from the recent literature cannot be ignored by the authors (see for instance, Maier et al. Chem. Eur. J. 20,1918 (2014) already cited but not discussed and Ciancaleoni et al. ACS Catal. 2015, 5, 803.)

Furthermore, the title insinuates for a key role of dispersive interactions while quantum mechanical calculations, in my view, point out for some weak and specific Au-substrate interactions (C-H...Au or Au π attraction).

Why is the Au₃-(2) complex more stable than Au₃(1)?

I would expect a more important stabilization for the latter (longer alkyl chain) if dispersive interactions were the driving-force.

A strictly related issue, from a methodological point of view, I feel uncomfortable that in the model employed

the DFT calculations water molecules are completely ignored.

If Au₃ has specific interactions with H protons of alkyl chain, one can expect even more specific interactions with water molecules, in this case one wonders if the stability order of the Au₃(3,1,2,4) complexes still holds.

In the current form the manuscript is unacceptable but promising.

Specific work is needed to make it acceptable:

- The discussion, in my view, would benefit of a significant review and rewriting, with the specific aim to put results in the context of the recent literature.

- DFT calculations need to be extended both including water molecules in the model and analyzing in details the metal substrate specific interactions (dispersive vs electrostatic...).

Furthermore, I suggest to work out

the reaction energy profile of the hydration reaction for all the four substrates, in order to make a more stringent and effective comparison with the experimental kinetic data.

Please find below our point-by-point reply to reviewers (our response in blue)

Reviewers' comments:

Reviewer #1 (Remarks to the Author):

This is a well-balanced study combining experiments, spectroscopic studies and a detailed computational study on very small gold clusters.

These sub-nanoparticles are a hot topic in catalysis research and enable very high turnover numbers.

The principle of stabilization of the sub-nanoparticles by an interaction of the alkyl side chains of substrates by dispersive interactions is a highly important one.

the manuscript clearly should be published in Nature communications.

Before the publication the following points have to be addressed:

"Recently, the use of exceedingly active small gold subnanoclusters (Au_n , $n < 10$) has been reported by the group of Corma (Science 338, 1452–1455 (2012))."

-- extend this sentence to

Instead of reference (1)

a specific review article should be provided, I would strongly suggest

"while Au(I) and Au(III) species are widely employed as homogeneous catalysts for the activation of alkenes and alkynes.1–8"

-- extend this to

"while Au(I) and Au(III) species are widely employed as homogeneous catalysts for the activation of alkenes, allenes and alkynes.1–8"

and add the key citation for allenes: Angew. Chem. Int. Ed. Engl. 2000, 39, 2285-2288.

"Most hydrogen bonds to gold atoms involve small, very polarized ligands^{19,20}"

-- here add a citation of Ito's pioneering milestone paper after polarized ligands:

J. Am. Chem. Soc. 1986, 108, 6405-6406

"also called subnanometric gold clusters or gold subnanoclusters.13–16 "

-- here as a reference add the detailed discussion of this success of Corma:

Science 2012, 338, 1434-1434.

"The scope of the Au-catalysed alkyne hydration"

-- after the word "hydratation" add a references:

"For high turnover numbers in the related spiroketalization of alkynes, see: Angew. Chem. Int. Ed. 2013, 52, 7963–7966"

"based on our previously proposed mechanism using Au(III) species as catalysts³⁸ (see Supp. Inf. Figs 3 and 4)."

-- here add two more references on details of these activations of alkynes by gold(III):

Schwerdtfeger's study in *Organometallics* 2010, 29, 2206–2210 and Pernpointner's fundamental relativistic study in *J. Chem. Theory Computation* 2009, 5, 2717–2725.

After these additions, the manuscript is ready

We thank very much the Reviewer for his very positive assessment of our work. Following his suggestions, we have included all the references suggested by him/her. In the case of the reference included in the manuscript abstract, we replaced the first paper that Corma *et al* published in this field with a more general article by the same authors (*Acc. Chem. Res.* 47, 834-844 (2014)) summarizing the reactivity of these subnanoclusters and comparing their reactivity with that of larger Au nanoparticles. The rest of references suggested by the Reviewer have been included in the Manuscript.

Reviewer #2 (Remarks to the Author):

This manuscript reports the preferential formation, stabilization and, incidentally, catalysis of ultrasmall Au clusters (<5 atoms) in the presence of long-chain alkyl alkynes. This phenomenon explains the different catalytic activity found for AuCl and AuPPh₃Cl as precursors of Au clusters during the hydration of different alkynes: while alkyl alkynes react well, phenylacetylene react sluggishly, a paradigmatic shift respect what should be expected (aromatic alkynes are more prone to nucleophilic addition) and also respect the well-known reactivity of phenylacetylene reported in the literature. Experimental and computational studies here support that the increasing number of Au—H-C dispersive interactions between the methylene groups of the alkyl chain of the alkyne and the ultrasmall Au clusters is the reason of their stabilization, and that when the cluster is >5 Au atoms, entropy takes over and more than one alkyne coordinates, thus Au—H-C interactions mainly disappear.

This is a foundational work that opens new ways of selectively when reacting alkynes. It seems plausible that the findings here shown are not only applicable to the hydration reaction but also to other hydroaddition reactions and other metal clusters, at least those of late heavy transition metals able to interact well with C-H bonds (Pd, Pt...). The manuscript is well-written and easy-to-follow, and the results seem easy to reproduce by other laboratories. Thus, I recommend publication after the following issues are addressed:

- Other nucleophiles rather than water: What about diols, amines and thiols? It would be nice to have some preliminary results, which would greatly improve the impact of the work.

We also thank the Reviewer for his/her supportive comments regarding the importance of our results and his/her constructive suggestions.

Regarding the use of other nucleophiles besides than water, we have followed his/her suggestions and have tested the reaction with amines, thiols, diols and long alkyl-chain alcohols as nucleophiles.

In the case of using propanethiol as a nucleophile with 1-hexyne and 2% mol AuCl as a catalyst, the formation of catalytic Au subnanoclusters is also precluded. Again, no conversion of the initial alkyne into the addition product was observed. Instead, stable Au(I) thiolate species $[\text{Au}_n(\text{propanethiolate})_{n+1}]^-$ were detected in the MALDI(-) spectrum.

Using propylamine as a nucleophile and 2% mol AuCl as a catalyst does not give the expected addition product to 1-hexyne even after stirring the reaction mixture for 22 h. A deeper analysis of the MALDI(+) spectrum after 5 h of reaction shows the formation of the corresponding Au(I) acetylide species $[\text{Au}_n(1\text{-hexyne-H})_{n-1}]^+$ ($n = 6, 7, 8$ and 9) that are stable enough to preclude the formation of subnanocluster species in the reaction media. The propylamine thus acts as a base in this process, promoting the deprotonation of 1-hexyne.

The corresponding MALDI(+) and MALDI(-) spectra have been added to the Supporting Information, and a short paragraph explaining these results has been included in the main text (page 8):

“Besides water and alcohols, other nucleophiles such as thiols and amines were tested. Addition of these nucleophiles to the tested alkynes was not observed in any case. Instead, peaks corresponding to Au(I) species were observed in the MALDI spectra of the reaction mixtures. With propanethiol, species of general formula $[Au_n(\text{propanethiolate})_{n+1}]^-$ were formed due to the higher acidity of thiols compared to alcohols, which precluded the formation of subnanoclusters. Similarly, peaks corresponding to $[Au_n(1\text{-hexyne-H})_{n-1}]^+$ ($n = 6, 7, 8$ and 9) species were observed in the MALDI+ spectrum when propylamine was used as a nucleophile; thus, under this conditions amines are able to deprotonate alkynes, again precluding the formation of Au_n subnanoclusters (see Supplementary Figure 2).”

- In alcohol media, it is probable that ketals are formed prior to the ketone (hydroalkoxylation reaction). Authors claim do not find ketals by GC-MS even at short reaction times, however, non-cyclic ketals will not be stable under the reaction conditions used. To detect if ketals are intermediates of the reaction, two possible methods can be employed: 1) Direct detection with a glycol as nucleophile, which forms the stable 5-member cyclic ketal; 2) As the alkyl chain increases, steric hindrance decreases the ketal formation rate, so the order of reactivity should be methanol>ethanol> isopentanol \approx pentanol>> decanol. This comparison must be done with the initial rate of formation of the ketone (the product is the same for all the alcohols) taken from kinetics; in the manuscript, authors refer to quantitative conversions in the alcohols but at final reaction time. A hydroalkoxylation reaction as first reaction step would explain the decreasing activity in 1-decanol.

Following the Reviewer’s suggestions, and connected to the previous point, we also tested the reaction between benzylacetylene and decanol in methanol, catalysed by 4% mol AuCl, aiming to decrease the potential ketal formation rate. Under these conditions, no reaction took place and 0% of the corresponding hydroalkoxylation product was observed, although minor amounts of the corresponding hydration product were detected due to the presence of traces of water in methanol.

However, the formation of a ketal intermediates was confirmed in one exceptional case (i.e. with 1,2-diols) following one of the Reviewer’s suggestions. Thus, the reaction between benzylacetylene and ethylene glycol (4% mol of AuCl) led to the quantitative formation of the corresponding diketal after 5 h. A short paragraph summarizing these results has been included in the main text (page 7).

“In an attempt to rule out potential hydroalkoxylation reactions taking place at short reaction times, yielding transient ketals that would ultimately hydrolysed to the observed ketones, long-chain alcohols such as 1-decanol in methanol were tested as nucleophiles in the presence of benzylacetylene and 4% mol AuCl as a catalyst. Under these conditions, no hydroxyalkylation products could be detected by GC-MS at any reaction time, and only trace amounts of the corresponding hydration product were observed due to the presence of traces of water in methanol. However, ketals could be observed when 1,2-diols were used as nucleophiles. As expected, the reaction between benzylacetylene and ethylene glycol in the presence of 4% mol of AuCl led to the quantitative formation of the corresponding cyclic ketal in 5 h”.

Also, we have included some recent references describing hydroalkoxylation reactions catalysed by gold species (references 34 to 36).

Reviewer #3 (Remarks to the Author):

The paper by Cordon et al. addresses the role of dispersive interactions in stabilizing small gold species in solution and the catalytic activity of small gold clusters in hydration reactions. The reviewer's comments are listed below.

1. The comparison of the performance of the catalyst needs additional support.

A) The overall kinetics plotted in Figure 1b for I Au and II Au shows very similar time dependency for the 2% samples. In the case of the 4% samples there are fewer data points shown for the II Au sample - this sample reaches ~95% conversion faster than the 4% I Au sample. Could the authors comment in more details and update the plots with error bars, if possible?

We thank the Reviewer for his/her suggestions, which significantly contributed to improve the quality of our manuscript.

We have repeated the time-resolved experiments of the II Au sample used (4% mol) in order to include additional intermediate points. Considering the new set of data, reaction is always faster (i.e. %conv is higher) for the I Au (4% mol) catalysed reaction. The corresponding plot in Figure 1b has been updated accordingly.

B) In order to further fortify the authors' claim the 4% I Au catalyst being the most active one, it would be very helpful if the authors could also include kinetics data obtained with the 2% and 4% II Au catalyst in reaction with some of the compounds (1) to (4).

Since we already demonstrated that AuCl is the most effective precatalyst for phenylacetylene hydration (Figure 1b), we did not repeat the time-resolved experiments with other substrates in order to avoid redundancy. Nevertheless, if the reviewer considers this point to be crucial, we can make an extra effort to perform such experiments.

2. The description of the NMR results shown in Supplemental Figure 1 should be expanded:

A) There is an apparent discrepancy: The figure caption mentions monitoring [Au(PPh₃)₂]Cl and AuCl, however the peaks in the plot are labeled as [Au(PPh₃)₂]⁺ and [ClAu(PPh₃)₂].

Considering the Reviewer suggestion, we have inspected and corrected Supplementary Figure 1 and its caption, which shows the evolution of catalyst [AuCl(PPh₃)] over time. As can be seen in the new version of the figure, after a short reaction time the presence of both [Au(PPh₃)₂]⁺ and [AuCl(PPh₃)] is apparent. This is an indirect evidence for the formation of AuCl, which is likely the precatalytic species from which Au_n subnanoclusters are formed. Supplementary Figures 1a and 1b have been corrected to better clarify this point.

B) The authors should also include a discussion of the evolution of the features with time and how this supports their claim made in the main text (page 5) about different rates of cluster formation when using different precursors.

A short paragraph explaining these observations has been included in the main text (page 5): “The slower production of subnanoclusters observed from the AuCl(PPh₃) precursor could be related to the slow formation of the real precatalyst AuCl through the equilibrium between AuCl(PPh₃) and AuCl and [Au(PPh₃)₂]Cl, as can be observed in the time-resolved ³¹P{¹H} NMR spectra (see Supp. Inf. Fig. 1)”.

3. A faster formation of Au clusters after the addition of HBF₄ is claimed through change of the reaction rate. Could the authors include direct evidence or a literature reference on the accelerated cluster growth under these conditions?

The only evidence we have for our claim is the observed more efficient conversion towards hydration when HBF₄ is used as a co-catalyst, similarly to what we observed in our previous report (ref. 37 in the new version of the manuscript). At this moment, we do not exactly know the role of Bronsted acids in this process. It could either be involved in the stabilization of the Au subnanoclusters, or in promoting the protodeauration step in the catalytic cycle. Since we could not find any valuable reference in the literature supporting our initial claim, and in order to avoid speculation, we decided to just report the experimental observation, but without any further claim on the role of HBF₄. Thus, the sentence reporting this result (page 5) now reads:

“In agreement with ~~the faster formation of Au_n clusters under acidic conditions~~, our previous observations,⁷⁸ the addition of HBF₄ (10% mol) led to the quantitative formation of acetophenone in 1h (with I) or 7.8 h (with II).”

4. MALDI-TOF is used to determine the size of the clusters (Supplemental Figure 2), which is a very elegant method to determine particle size.

A) In the caption of FigS2 Au₃ is described as the most abundant species at early reaction times, while the main text refers to the abundance of 1-5 atom clusters. For clarity, can the authors assign all peaks in the spectra?

Regarding the MALDI-TOF spectra in Supplementary Figures 3 and 4, we missed including the spectrum at the lowest *m/z* values, which shows the peaks corresponding to Au₁ and Au₂ species. Indeed, at early stages of the reaction, additional peaks corresponding to [AuCl(phenylacetylide)]⁻ (*m/z* = 333), [Au₂Cl₂(phenylacetylide)]⁻ (*m/z* = 565), [Au₂Cl(phenylacetylide)₂]⁻ (*m/z* = 630), [Au₂(C≡CPh)₃]⁻ (*m/z* = 697), [Au₃Cl₂(C≡CPh)₂]⁻ (*m/z* = 865), [Au₃Cl(C≡CPh)₃]⁻ (*m/z* = 929) and [Au₃(C≡CPh)₄]⁻ (*m/z* = 995), among other peaks, also appear. The presence of these species is expected at initial reaction times when the precatalyst AuCl is being formed, and the catalytic clusters have not been completely deactivated. To further clarify this point, new and more detailed Supplementary Figures 3 and 4 with labelled peaks has been created.

B) For a better assessment of the size of clusters present at early and late times, could the authors expand both spectra for showing a range corresponding to cluster sizes ~1-20?

In MALDI experiments, different m/z ranges are analysed in order to optimize the detector performance. Also, peak intensities are not comparable across the different m/z ranges, since different laser powers are needed. Nevertheless, in the case of the hydration of phenylacetylene using 2% mol of AuCl as a precatalyst we include the spectra showing the high m/z range, registered at short reaction times (55 min) in order to show up the absence of large Au subnanoclusters (Supplementary Figure 3 (bottom)), which are indeed observed at longer reaction times (262 h) (Supplementary Figure 4).

5. Page 5 of the main text refers to solvent effects to be listed in Supplementary Table 2. However, the table lists kinetics data for different substrates which are referred to on page 6 of the manuscript.

We have deleted the reference to solvent effects pointing to Table 2 in Supplementary Material since these results are already included in the main text on page 5.

6. Photoluminescence is used to estimate cluster sizes, based on literature references.

A) This reviewer is not a specialist and would like to learn from the paper how to distinguish size effects from ligand/solvent effects. Could the authors discuss this point in the manuscript?

We thank again the Reviewer for his/her comment, which has helped to improving the clarity of this part of the manuscript. Regarding the question about the role of the size or the solvent/ligand effect on the emission energies obtained for the different $Au_n(\text{alkyne})_m$ species, several issues should be considered: (i) According to previous reports on the luminescence of dendrimer encapsulated Au_n subnanoclusters (ref. 42 in the main text), we can assume that the size of the Au_n subnanoclusters increase the emission shifts to lower energy; (ii) as the Reviewer suggests, solvent effects always take place in solution experiments. This effect roughly decreases the energy of the excited state as the solvent polarity increases; nevertheless, in our case we can rule out such a solvent effect since all the experiments are carried out in the same solvent (methanol); (iii) Also, as the Reviewer points out, emitting species surrounded by interacting ligands always produce a change on the energy of the molecular orbitals involved in the electronic transition responsible for the luminescent behaviour. In order to the relevance of this effect in our system, we performed DFT calculations on the electronic structure of model systems [$Au_3(1\text{-hexyne})$] and [$Au_6(1\text{-hexyne})_2$] and computing their corresponding HOMO and LUMO orbitals. As can be seen in the new Supplementary Figure 5, a small contribution from the alkyne ligand compared to that from the Au_n subnanocluster was computed for both model systems. Therefore, the alkyne ligands may have some, albeit not significant, influence on the emission energy of the $Au_n(\text{alkyne})_m$ species; (iv) However, the experimental observation of a low energy emission being produced when phenylacetylene is used, and that higher and similar energy emissions are produced with 1-hexyne, 1-dodecyne or benzylacetylene, suggests that the subnanocluster size is the most

important parameter determining the emission energy. This is in agreement with the previously reported results on size-dependent Au_n subnanoclusters luminescence (references 39-41 in the main text). Therefore, and following the Reviewer's suggestion, we have included the next paragraph in the main text further to clarify this point (page 8):

“Besides particle size, other factors potentially affecting the emission energies in these systems could be the ligand(s) and/or the solvent interacting with the subnanoclusters. Solvent effects, despite having an influence of the energy of the excited states, can be safely ruled out in our case because all the experiments are carried out in the same solvent; we assume that solvent effects are independent of the subnanocluster size. The influence of the alkyne ligands on the molecular orbitals involved in the electronic transition responsible for the observed luminescence, was evaluated through DFT calculations on model systems $[Au_3(1\text{-hexyne})]$ and $[Au_6(1\text{-hexyne})_2]$. A very small contribution from the alkyne ligand compared to that from the Au_n subnanocluster was observed on the computed HOMO and LUMO orbitals (see Supplementary Figure 5). Therefore, we can conclude that the coordinated alkynes may have some, although not decisive, influence on the emission energy of the $Au_n(\text{alkyne})_m$ species.”

B) The time resolved emission spectra show a very nice correlation between catalytic conversion and abundance of the species emitting at the given wavelength. Did the authors also collect emission data at other wavelengths to monitor the presence of different size clusters?

We thank again the Reviewer for his/her interesting suggestion. We have repeated the measurements for the hydration of 1-dodecyne, collecting the emission spectra by exciting at different wavelengths at the same reaction time in order to check the existence of emitting species of larger sizes. The only emission detected appears at 515 nm (excitation between 290-350 nm) and corresponds to small size subnanoclusters interacting with 1-dodecyne. When exciting at 400 nm looking for larger subnanoclusters, no emission was detected. Conversely, a low energy emission at 601 nm (excitation at 400 nm) is detected with phenylacetylene, confirming the presence of large size subnanoclusters.

Other comments:

- This reviewer also suggests adding more details on MALDI characterization in the Methods section, including sample handling.

As the Reviewer suggests we have included the sample handling information for the MALDI experiments in the Online Methods:

“The MALDI-TOF samples were prepared by adding 1 μ L of the reaction mixture on a spot of a MSP ground steel BC sample holder (Bruker). The samples were air dried before measurement.”

- The font size is very small in some of the figures.

This issue has been corrected in the new version of the manuscript when possible.

In summary, Cordon et al. amassed an impressive set of data. However, in the opinion of this reviewer, a more focused presentation would have been better suited for a short communication. This reviewer feels that this work is better positioned for a full paper in a more specialized journal.

We truly acknowledge the Reviewer's kind words and hope that our work is suited for a high-impact, broad audience journal such as Nature Communications.

Reviewer #4 (Remarks to the Author):

The work may be in principle of interest for the wide chemists community working in homogeneous and heterogeneous catalysis. The authors claim that the transient stabilization of very small gold subnanoparticle (for example Au₃) can be achieved appending alkyl chains or aromatic group to the coordinating π bond of simple alkynes.

The authors tended to correlate the superior thermodynamic stability of small gold cluster with alkynes bearing alkyl chains or aromatic groups with experimental kinetic-profiles for alkynes hydration reaction. This correlation is interesting but, in my view, authors do not appropriately discuss their results in the context of previous literature.

The catalyst stability is clearly a key point, however pre-equilibrium, nucleophilic addition, and protodeauration are as much as important and the final outcomes depend on each of these steps.

Several works in the literature have already pointed out most of these aspects just as in the case of hydroalkoxylation of alkynes mediated by gold.

The interpretative framework emerging from the recent literature cannot be ignored by the authors (see for instance, Maier et al. *Chem. Eur. J.* **20**,1918 (2014) already cited but not discussed and Ciancaleoni et al. *ACS Catal.* **2015**, *5*, 803.)

We acknowledge the Reviewer's concern about not thoroughly discussing the detailed reaction mechanism in our manuscript, in the context of previous literature. In fact, we already discussed the importance of the nucleophilic addition and protodeauration steps in our previous paper, also describing the hydration of alkynes but using Gold(III) complexes as catalysts (*Organometallics* **33**, 3823–3830 (2014), ref. 37 in the paper). The present manuscript, although includes the computation of the whole reaction profile(s), is much more focused on the formation and stability of small subnanoclusters, which we believe is the catalytic bottleneck for the described reaction. On the other hand, the very good references pointed out by the Reviewer are focused on a similar, but not the same reaction, namely hydroxyalkylation, using completely different catalysts such as Au(I) species, and describing the effect of diaurated species and counteranion on the mechanism. While we consider these publications highly valuable, we cannot find a proper way of connecting our present work with them.

Attending the Reviewer's request, we included the following reference as ref. 36:

“Ciancaleoni, G., Belpassi, L., Zuccaccia, D., Tarantelli, F. & Belanzoni, P. Counterion Effect in the Reaction Mechanism of NHC Gold(I)-Catalyzed Alkoxylation of Alkynes: Computational Insight into Experiment. ACS Catalysis 5, 803–814 (2015)”

Furthermore, the title insinuates for a key role of dispersive interactions while quantum mechanical calculations, in my view, point out for some weak and specific Au-substrate interactions (C-H..Au or Au π attraction).

Dispersion interactions (instantaneous dipole-induced dipole forces), are indeed involved in the kind of interactions we describe in our paper. Gold subnanoclusters are neutral, big-sized species and thus, the contribution of dispersion to the strength of C-H..Au and π ..Au bonds can be significant, sometimes comparable to electrostatic and polarization contributions. To further support this idea, we have performed a comparative Morokuma-type Energy Decomposition Analysis of simple models, namely *Au₃-acetylene*, *Au₃-benzene*, *Au₃-water* and *Au₃-butane*. This analysis reveals that, while *Au₃-acetylene*, *Au₃-benzene* and *Au₃-water* interactions are clearly dominated by electrostatic/polarization attractions (56%, 48% and 53% of the total attractive interaction energy, versus 10%, 18% and 15% of dispersion, respectively), in the case of *Au₃-butane*, dispersion (34%) is equally important to electrostatic/polarization (34%). Thus, since a *Au₃-acetylene*-like interaction is always present in all the considered catalytic systems, we do believe that dispersive C-H bonds have a decisive contribution to stabilize small Au subnanoclusters when alkynes such as 1-hexyne and 1-dodecyne are used.

We have summarized this information in the new Supplementary Figure 11 and added the following sentences to the *Computational details* section of the Online Methods:

“Diverse contributions (electrostatic, exchange, repulsion, polarization and dispersion) to the interaction energies were calculated for selected model Au₃ adducts through the localized molecular orbital energy decomposition analysis (LMO-EDA)⁶² as implemented in Gamess.⁶³ This analysis revealed that, while Au₃-acetylene, Au₃-benzene and Au₃-water interactions are clearly dominated by electrostatic/polarization attractions, in the case of Au₃-butane, dispersion is equally important to electrostatic/polarization (Supplementary Figure 11).”

Nevertheless, and in order to address the Reviewer’s concern, we have removed the word “Dispersive” from the title”, and just mention, as the Reviewer suggests, “Au-Substrate Interactions”.

Why is the Au₃(2) complex more stable than Au₃(1)? I would expect a more important stabilization for the latter (longer alkyl chain) if dispersive interactions were the driving-force.

The Reviewer is right, complexes of Au₃ with 1-dodecyne (bearing a longer alkyl chain) would be, in principle, expected to be more stabilized than with shorter 1-hexyne from a purely enthalpic point of view. However, it must be taken into account that longer alkyl chains suffer a larger conformational penalty when they fold from extended conformations to folded ones, such as in the case of 1-dodecyne. This is clearly reflected from the energies of the linear and folded conformations of 1-hexyne ($\Delta G_{\text{folded-linear}} = 0.0 \text{ kcal mol}^{-1}$) and 1-dodecyne ($\Delta G_{\text{folded-linear}} = +2.9 \text{ kcal mol}^{-1}$). This difference in “folding energy” is in line with the superior stability of the Au₃-(1-hexyne) complexes.

In order to further clarify this point, we have added the following paragraph to page 11:

“Note that Au₃-(2), despite achieving a larger number of Au···H–C interactions, shows a smaller interaction energy than Au₃-(1) and Au₃-(3). This is due to the conformational penalty ($\Delta G_{\text{folded-linear}} = +2.9 \text{ kcal mol}^{-1}$) experienced by 1-dodecyne (2) upon folding into the conformation that is recognized by the Au₃ subnanocluster; such conformational penalty is negligible for less flexible 1-hexyne (1) ($\Delta G_{\text{folded-linear}} = +0.0 \text{ kcal mol}^{-1}$) (see Supplementary Table 3).”

We have also added the thermodynamic data calculated for free alkynes 1-4 to Supplementary Table 3.

A strictly related issue, from a methodological point of view, I feel uncomfortable that in the model employed the DFT calculations water molecules are completely ignored.

If Au₃ has specific interactions with H protons of alkyl chain, one can expect even more specific interactions with water molecules, in this case one wonders if the stability order of the Au₃(3,1,2,4) complexes still holds.

We agree with the Reviewer that the role of explicit solvation should be examined, although this issue is commonly overlooked in theoretical studies on catalytic processes. Indeed, we did consider coordination of solvent (methanol) to Au₃ in the previous version of our manuscript, when we compared the whole hydration profile of phenylacetylene catalysed by either naked Au₃ or Au₃(MeOH)₃ (Supplementary Figure 6). From these calculations, it was apparent that explicit solvation did not change significantly the energetics of the reaction, apart from making coordination of the alkyne slightly endergonic.

Attending the Reviewer’s request, we calculated the complexation energies of differently solvated Au₃ species with alkynes **1-4**, in order to evaluate the influence of solvent on the stability of each Au₃-alkyne complex.

It should be noted that, with such complicated reaction mixtures –methanol and water being the major and minor components, respectively–, it is not trivial to decide which solvated species should be considered for discussion. Au₃ could be coordinated, in principle to: 3 methanol, 2 methanol + 1 water, 1 methanol + 2 water, 3 water molecules...and in reality an statistical distribution of all these species will surely exist. Moreover, we found that when 2 methanol/water molecules are coordinated to Au₃ through their more coordinating oxygen atom (Au···OH), the electronic structure of the cluster is perturbed in such a way that the coordination of a third ligand (either solvent or the reacting alkyne) becomes quite unfavorable, and sometimes solvent coordinates through its polarized hydrogen (Au···HO), further complicating the picture. Thus, it is energetically more favorable and computationally more practical to keep Au₃ monosolvated; this leaves enough space and empty orbitals to allow the necessary coordination of the reacting alkyne without imposing a too high energy penalty.

Nevertheless, and answering the Reviewer’s question, irrespective of the nature (methanol or water) or number of solvent molecules coordinated to Au₃, and the energetic penalty associated to the process, the general trend calculated for the unsolvated Au₃ clusters is maintained: complexes with phenylacetylene (**1**), which lacks a flexible sidechain able to interact with the metal cluster, are always less stable than those with 1-hexyne (**1**), 1-dodecyne (**2**) and benzylacetylene (**3**).

We have summarized this information in the new Supplementary Figure 9 and added the following paragraph to page 13 to further clarify this point:

“The role of solvent on the relative stability of Au₃-alkyne complexes was examined (Supplementary Figure 9). Irrespective of the nature (methanol or water) or number of solvent molecules coordinated to Au₃, the general trend calculated for the unsolvated Au₃ clusters is maintained: complexes with phenylacetylene (1), which lacks a flexible sidechain able to interact with the metal cluster, are always less stable than those with 1-hexyne (1), 1-dodecyne (2) and benzylacetylene (3).”

In the current form the manuscript is unacceptable but promising.

Specific work is needed to make it acceptable:

- The discussion, in my view, would benefit of a significant review and rewriting, with the specific aim to put results in the context of the recent literature.

Please see comments above on this regard.

- DFT calculations need to be extended both including water molecules in the model and analyzing in details the metal substrate specific interactions (dispersive vs electrostatic...).

Please see comments above on this regard.

Furthermore, I suggest to work out the reaction energy profile of the hydration reaction for all the four substrates, in order to make a more stringent and effective comparison with the experimental kinetic data.

We thank the Reviewer for this very appropriate suggestion. We have recalculated the whole free energy profile for the Au₃-catalyzed hydration of the most reactive alkyne (*i.e.* 1-hexyne **1**) and added the results to an updated version of Supplementary Figure 8. While the enol-to-keto tautomerization (ts5) and protodeauration steps (ts7) have similar activation barriers than those calculated for phenylacetylene, the initial nucleophilic attack (ts3) to Au₃-**1** has a higher activation barrier than to Au₃-**4**, as expected, due to the higher stability of the Au₃ complex with 1-hexyne and the poorer electrophilic character of 1-hexyne compared to phenylacetylene. These results strongly suggest that the formation and stability of the smallest Au subnanoclusters is the catalytic bottleneck for this process, given that a kinetically less reactive alkyne (1-hexyne) is in fact the one that shows the highest hydration rates.

Also, the superior stability of the Au₃-**1** complexes towards aggregation is reflected by the fact that we could not find any transition state or intermediate for the linearization of the Au₃ cluster starting from this adduct, which could be considered a prerequisite for the formation of larger clusters such as Au₆. Any attempt to linearize Au₃ in Au₃-**1** quickly rearranged back to the

triangular ground state, reflecting the protecting properties of alkynes bearing medium to long alkyl/aryl sidechains. However, such Au₃ linearization transition state and intermediates could be in fact found for Au₃-(**4**), suggesting its higher propensity to achieve higher nuclearity clusters.

We have included the following paragraphs in pages 11 and 13 to include this information:

*“The whole free energy profile for the Au₃-catalyzed hydration of the experimentally most reactive alkyne, i.e. 1-hexyne (**1**) was subsequently calculated (Supplementary Figure 8). While the enol-to-keto tautomerization (ts5) and protodeauration steps (ts7) have similar activation barriers than those calculated for phenylacetylene (**4**), the initial nucleophilic attack (ts3) to Au₃-(**1**) has a higher activation barrier than to Au₃-(**4**), due to the aforementioned higher stability of the Au₃ complex with 1-hexyne (**1**) and the poorer electrophilic character of 1-hexyne (**1**) compared to phenylacetylene (**4**). These results strongly suggest that the formation and stability of the smallest Au subnanoclusters is indeed the catalytic bottleneck for this process, given that a kinetically less reactive alkyne such as 1-hexyne (**1**) is in fact the one that shows the highest hydration rates..”*

*“Finally, the superior stability of the Au₃-(**1**) complexes towards aggregation is reflected by the fact that we could not find any transition state or intermediate for the linearization of the Au₃ cluster starting from this adduct, which could be envisioned as a prerequisite for the formation of larger clusters such as Au₆. Any attempt to linearize Au₃ in Au₃-(**1**) quickly rearranged back to the triangular ground state, reflecting the protecting properties of alkynes bearing medium to long alkyl/aryl sidechains. However, such Au₃ linearization transition state and intermediates could be indeed found for Au₃-(**4**) (Supplementary Table 3), suggesting its higher propensity to achieve higher nuclearity clusters.”*

Reviewer #1 (Remarks to the Author):

Authors have fulfilled satisfactorily the issues arised during the review, the manuscript is ready for publication.

Reviewer #2 (Remarks to the Author):

In their revision the authors acceptably addressed the comments and inquiries of this referee.

Reviewer #3 (Remarks to the Author):

As I can judge the manuscript is now significantly improved and most of my doubts have been fully addressed. This is a nice peace of work that deserves publication in Nature Communication.